# A human antibody selective for transthyretin amyloid removes cardiac amyloid through phagocytic immune cells

Aubin Michalon [1✉], Andreas Hagenbuch[1], Christian Huy[1], Evita Varela [1], Benoit Combaluzier[1], Thibaud Damy[2], Ole B. Suhr [3], Maria J. Saraiva [4], Christoph Hock[1,5], Roger M. Nitsch[1,5] & Jan Grimm [1]

Transthyretin amyloid (ATTR) cardiomyopathy is a debilitating disease leading to heart failure and death. It is characterized by the deposition of extracellular ATTR fibrils in the myocardium. Reducing myocardial ATTR load is a therapeutic goal anticipated to translate into restored cardiac function and improved patient survival. For this purpose, we developed the selective anti-ATTR antibody NI301A, a recombinant human monoclonal immunoglobulin G1. NI301A was cloned following comprehensive analyses of memory B cell repertoires derived from healthy elderly subjects. NI301A binds selectively with high affinity to the disease-associated ATTR aggregates of either wild-type or variant ATTR related to sporadic or hereditary disease, respectively. It does not bind physiological transthyretin. NI301A removes ATTR deposits ex vivo from patient-derived myocardium by macrophages, as well as in vivo from mice grafted with patient-derived ATTR fibrils in a dose- and time-dependent fashion. The biological activity of ATTR removal involves antibody-mediated activation of phagocytic immune cells including macrophages. These data support the evaluation of safety and tolerability of NI301A in an ongoing phase 1 clinical trial in patients with ATTR cardiomyopathy.

[1] Neurimmune, Schlieren, Switzerland. [2] Referral Center for Cardiac Amyloidosis and Department of Cardiology, Henri Mondor University Hospital, Créteil, France. [3] Department of Public Health and Clinical Medicine, Umea University, Umea, Sweden. [4] i3S - Instituto de Investigação e Inovação em Saúde & IBMC - Instituto de Biologia Molecular e Celular, Porto, Portugal. [5] Institute for Regenerative Medicine (IREM), University of Zurich, Zurich, Switzerland. ✉email: aubin.michalon@neurimmune.com

Transthyretin (TTR) amyloidosis is a systemic disease characterized by the accumulation of TTR amyloid (ATTR) in myocardium, peripheral nerves, and various other tissues, causing congestive heart failure, polyneuropathy, and death[1]. Sporadic and hereditary forms of the disease are characterized by the deposition of wild-type TTR (TTRwt) and mutated variants of TTR (TTRv), respectively. TTR is a hepatic thyroxine and retinol-transporting protein secreted into the blood[2]. It can partially unfold and aggregate into ATTR fibrils; these consists of stable β-pleated sheet structures[3]. Extracellular myocardial ATTR deposits accumulate in massive amounts impacting such mechanical properties of the myocardium as diastolic relaxation followed by systolic impairments during progression of the disease. In peripheral sensorimotor neurons, ATTR deposits are causing polyneuropathy, which are, together with cardiomyopathy, the major clinical manifestations of TTR amyloidosis[4]. Most sporadic and many genetic amyloidosis are associated with both cardiomyopathy and polyneuropathy. Rare pure forms of either syndrome include the TTR V122I variant with dominating cardiomyopathy and the TTR-V30M variant predominantly associated with polyneuropathy[5]. Epidemiological studies indicate that TTR amyloidosis with cardiomyopathy (ATTR-CM) is likely massively underdiagnosed. Data from Finland, USA, Spain, and France consistently indicate that cardiac TTR amyloidosis causes 5–13% of all heart failures with preserved ejection fraction or hypertrophic cardiomyopathies[6–11].

Current treatments of TTR amyloidosis include stabilization of TTR in its naturally folded physiological form and gene silencing, thus preventing unfolding, aggregation, and deposition in target tissues[12]. To remove pre-existing ATTR from affected tissues in patients with TTR amyloidosis, we developed the human monoclonal antibody NI301A selectively targeting ATTR aggregates. By using high-throughput screening of human memory B-cell libraries, we generated a collection of ATTR-binding antibodies selected for high-affinity binding to ATTR, amyloid-removal activity, and absent binding to naturally folded, physiological TTR. This functional selection process led to the identification of NI301A. NI301A binds to the linear epitope WEPFA hidden in TTR's naturally folded conformation, but accessible to antibody binding following unfolding and aggregation. NI301A binds ATTR deposits in cardiac tissues obtained at autopsy from ATTR-CM patients and triggers ATTR removal from the post-mortem tissues ex vivo via phagocytosis through added human macrophages. In wild-type mice grafted with patient-derived ATTR fibrils, systemically administered NI301A accumulates on the grafts and rapidly removes them via Fcγ-mediated phagocytosis. Together, these data support the clinical testing of NI301A (NI006) in patients with ATTR-CM in an ongoing phase 1 clinical trial (NCT04360434).

## Results

### NI301A selectively binds human ATTR. NI301A bound with high affinity and specificity both ATTRwt and ATTRv aggregates, without binding to their native physiological conformation.

In enzyme-linked immunosorbent assays (ELISAs) against human ATTR fibrils prepared in vitro (Fig. 1a) and coated to assay plates, NI301A bound ATTRwt, as well as ATTR-V30M and ATTR-V122I, with high affinities as indicated by $EC_{50}$ values in the sub-nanomolar range ($EC_{50}$ for ATTRwt = 0.35 nM, ATTR-V30M = 0.38 nM, and ATTR-V122I = 0.15 nM; Fig. 1b). NI301A bound similarly to fibrils of ATTR-V20I, L55P, S112I, and S116Y ($EC_{50}$ for ATTR-V20I = 0.14 nM, ATTR-L55P = 0.61 nM, ATTR-S112I = 0.20 nM, and ATTR-S116Y = 0.24 nM; Fig. 1b).

To confirm aggregate selectivity independently of high-density coating, fibrils were presented in solution by using a sandwich ELISA format. Again, NI301A bound with high affinity to ATTR oligomers with $EC_{50}$ values in the low nanomolar range ($EC_{50}$ for ATTRwt = 3.7 nM, ATTR-V30M = 12.8 nM and for ATTR-V122I = 3.7 nM). NI301A did not bind to native TTR tetramers in their physiological conformation (Fig. 1c).

Surface plasmon resonance (SPR) analyses of binding kinetics showed that NI301A bound ATTRwt oligomers in solution with an equilibrium dissociation constant ($K_D$) of 1.2 nM, with rapid association and slow dissociation ($k_a = 2.1 \times 10^4$ $M^{-1}s^{-1}$, $k_d = 2.6 \times 10^{-5}$ $s^{-1}$; Fig. 1d). Biolayer interferometry (BLI) yielded similar results for ATTRv: NI301A bound ATTR-V30M oligomers in solution with a $K_D$ of 2.5 nM ($k_a = 1.7 \times 10^4$ $M^{-1}s^{-1}$, $k_d = 4.3 \times 10^{-5}$ $s^{-1}$) and ATTR-V122I oligomers with a $K_D$ of 3.6 nM ($k_a = 2.2 \times 10^4$ $M^{-1}s^{-1}$, $k_d = 7.8 \times 10^{-5}$ $s^{-1}$; Fig. 1e). Consistent with its conformation selectivity indicated by sandwich ELISA, NI301A did not bind physiological native tetramers of either TTRwt or TTRv.

NI301A's selectivity for ATTR was further confirmed by dot blot analyses: NI301A bound fibrils of ATTRwt and ATTR-V20I, V30M, L55P, S77Y, S112I, Y116S, T119M, and V122I immobilized on nitrocellulose membranes, but not the cognate native TTR tetramers (Fig. 1f). In contrast, a non-selective control anti-TTR antiserum (Dako A0002) bound both ATTR fibrils and native TTR tetramers.

Using overlapping peptide arrays, the NI301A-binding epitope was determined as the linear peptide sequence WEPFA located in position 41–45 of mature TTR protein (Supplementary Fig. 1).

Time-course aggregation studies of in vitro-generated ATTR-V30M fibrils separated by semi-native polyacrylamide gel electrophoresis (PAGE) characterized NI301A binding to specific oligomeric ATTR species: NI301A bound dimers, trimers, tetramers, as well as higher-order ATTR aggregates (Fig. 1g). As expected, NI301A did not react with native TTR tetramers but detected misfolded ATTR monomers exposing the NI301A-binding epitope that is hidden in the native monomer. In contrast, the non-selective control antiserum Dako A0002 bound both the misfolded monomer and the native tetramer. The antibody 39-44, described as an amyloid-selective TTR antibody[13], robustly bound dimers and small ATTR oligomers but only poorly to larger ATTR oligomers with residual binding to native TTR tetramers.

The absence of binding to physiological TTR tetramers was further investigated in a series of immunoprecipitation experiments. Whereas Dako A0002 immunoprecipitated native TTR from human plasma even at 1 : 10,000 dilution, NI301A failed to immunoprecipitate TTR from the plasma (Fig. 1h and Supplementary Fig. 2).

### NI301A binds ATTR deposits on patient tissues. Immunohistochemistry (IHC) showed that NI301A bound with high-selectivity ATTR deposits on a collection of fresh frozen cardiac tissues obtained at autopsy from patients with sporadic ATTR-CM (Fig. 2a). Consistent with its high selectivity for ATTR, NI301A failed to bind to cardiac non-ATTR deposits (A+TTR−), likely representing light-chain amyloid.

IHC staining with NI301A of fresh frozen salivary gland biopsies collected from ATTR T49I, T60A, I68L, S77T, or V122I mutation carriers showed a pronounced staining of ATTR deposits in all patients analyzed (Fig. 2b). Furthermore, NI301A bound ATTR deposits in skin biopsies and abdominal fat aspirates collected from hereditary cases of ATTR-V30M amyloidosis with polyneuropathy (Fig. 2c, d). In contrast to the non-selective anti-TTR antiserum Dako A0002, NI301A did not bind to native TTR expressed in pancreatic α-cells (Fig. 2c), confirming further the amyloid selectivity of NI301A.

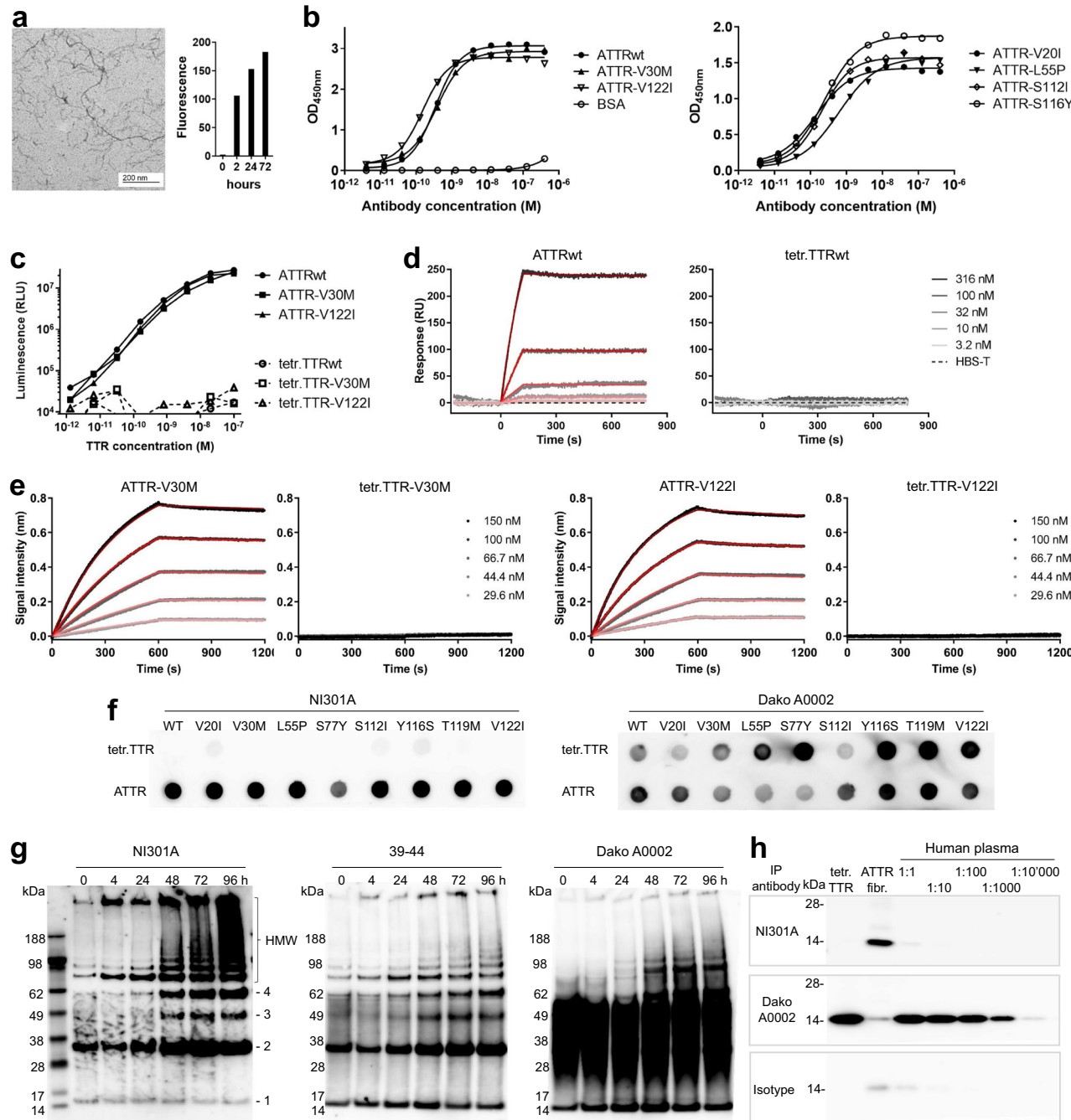

**Fig. 1 NI301A binds to wild-type and mutant ATTR fibrils in vitro, with absent binding to native TTR tetramers. a** Characterization of ATTRwt fibrils prepared in vitro: fibrils were visualized by electron microscopy after 72 h and bound the amyloid indicator dye thioflavin T. **b** NI301A binding to coated ATTRwt and ATTRv fibrils by ELISA (symbols: mean of duplicates, lines: data fitting). **c** Sandwich ELISA using NI301A for capture and detection to measure in-solution binding to ATTR and TTR tetramers (mean of duplicates). **d** SPR analysis of NI301A-binding kinetics to ATTRwt oligomers and native TTRwt tetramers in solution. **e** BLI analysis of NI301A-binding kinetics to ATTR-V30M and ATTR-V122I oligomers, and native TTRv tetramers in solution. For **d** and **e**, data (gray to black) were fitted with a 1 : 1 binding model (red lines). **f** Dot blot analysis of NI301A binding to native tetramers or amyloid aggregates of TTRwt and TTRv proteins. A polyclonal, pan-TTR antibody (Dako A0002) was used for comparison. **g** Analysis of TTR-V30M aggregation time course by semi-native SDS-PAGE followed by western blotting with antibodies NI301A, 39-44, and Dako A0002. Positions of ATTR monomers (1), dimers (2), trimers (3), tetramers (4), and aggregates of high molecular weight (HMW) are marked. **h** Immunoprecipitation with antibodies NI301A, Dako A0002, and human IgG1 isotype control of native TTR tetramers (tetr.TTR) or patient-derived ATTR fibrils (ATTR fibr.), and serial dilutions of human plasma from healthy donors.

On patient myocardium amyloid extracts, NI301A reacted with high sensitivity to small, medium, and large ATTR aggregates, consistent with our results with ATTR-V30M aggregates prepared in vitro. Consistent with its ATTR selectivity, NI301A did not bind corresponding preparations from cardiac tissues without amyloid deposits (A−) or amyloid deposits unrelated to TTR (A+TTR−) (Fig. 3). Selective binding to ATTR aggregates together with absent binding to unrelated proteins was sustained following long exposure, further underscoring NI301A's high degree of selectivity (Supplementary Fig. 3).

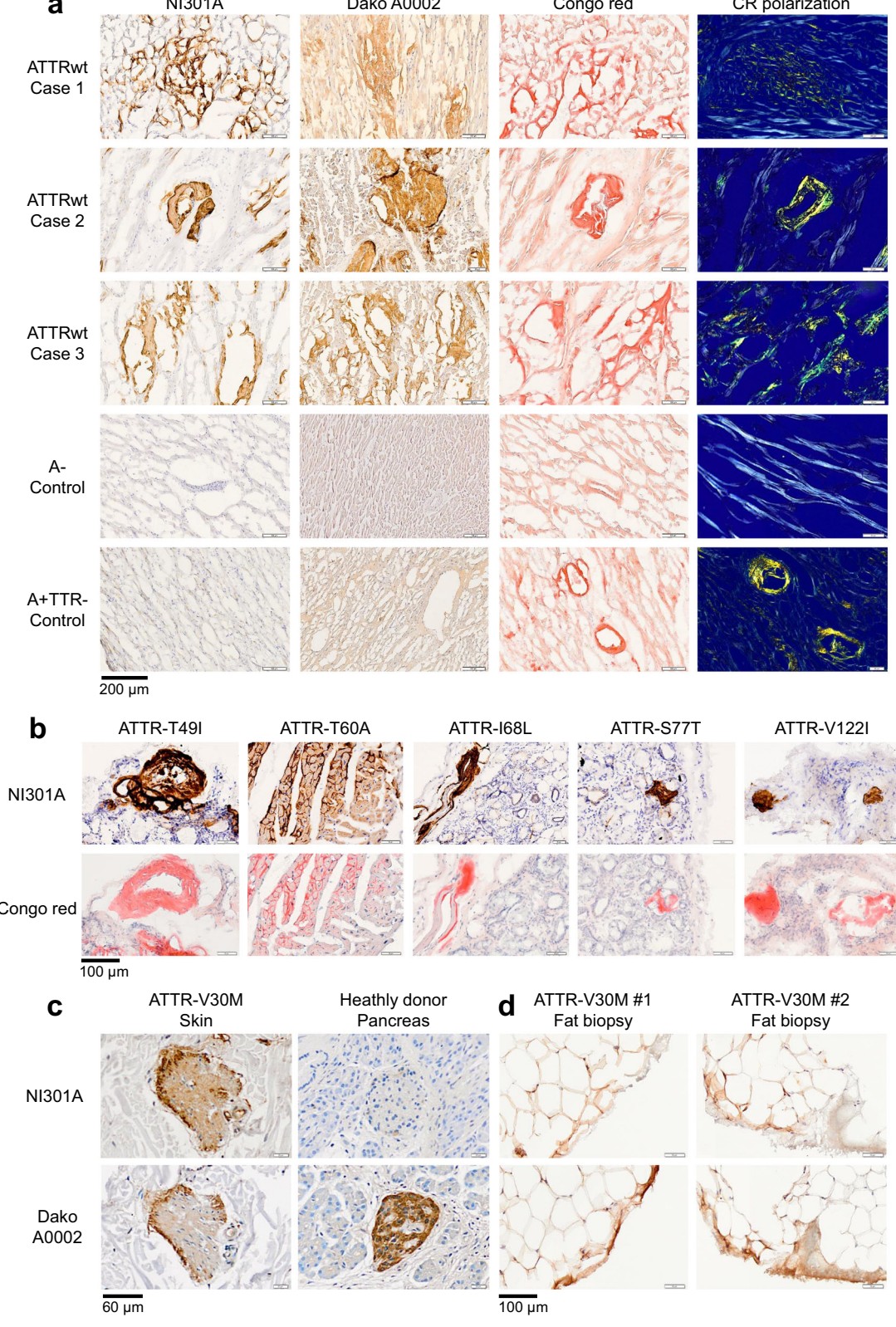

**Fig. 2 NI301A binds TTR amyloid present in sporadic and hereditary cases of ATTR amyloidosis with cardiomyopathy and polyneuropathy. a** IHC using NI301A on frozen post-mortem heart tissues collected from three sporadic cases with ATTRwt cardiomyopathy, a case without amyloid (A−), and one with non-TTR amyloid deposits (A+TTR−). ATTR deposits were identified by Congo red staining with birefringence and immunostaining with the pan-TTR antibody Dako A0002. **b** IHC using NI301A and Congo red staining on frozen salivary gland biopsies collected from ATTRv cardiomyopathy patients carrying the variants TTR-T49I, T60A, I68L, S77T, and V122I. **c** IHC using NI301A and Dako A0002 on skin biopsy from a patient with ATTR-V30M polyneuropathy and on human pancreas tissue. **d** IHC using NI301A and Dako A0002 on abdominal fat aspirates from two patients with ATTR-V30M polyneuropathy.

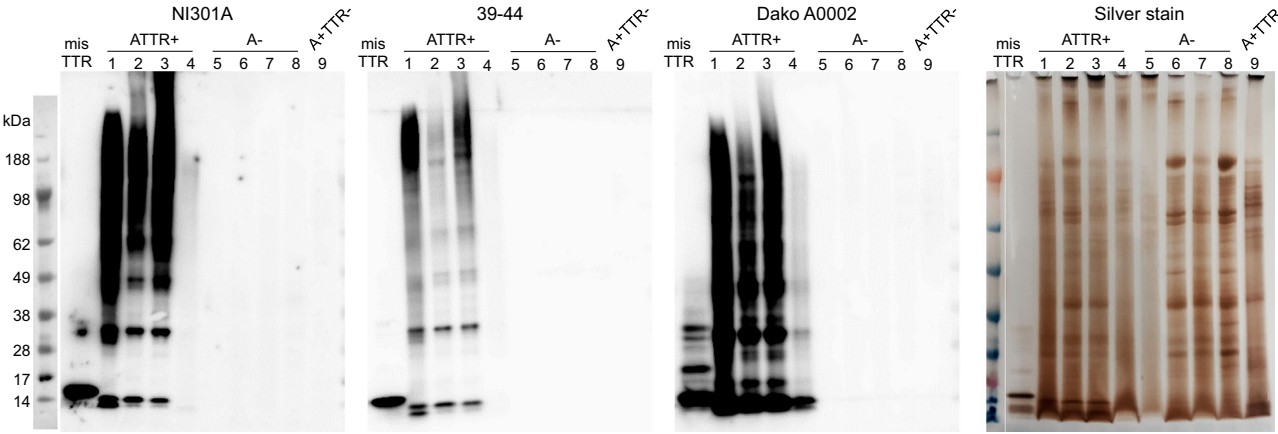

**Fig. 3 NI301A binds small and large ATTRwt aggregates extracted from patient tissues.** Semi-native SDS-PAGE analysis of the amyloid fibril extracts prepared from frozen post-mortem cardiac tissues presenting TTR amyloid (ATTR+ samples, #1–4), no amyloid (A− samples, #5–8), or non-TTR amyloid (A+TTR− sample, # 9). Identical SDS-PAGE gels were processed for western blot analysis with the antibodies NI301A, 39-44, and Dako A0002, or stained with the silver method. The low ATTR staining intensity of sample 4 is consistent with the low amount of TTR amyloid observed by IHC in this sample.

The selectivity of NI301A was further confirmed in a human tissue cross-reactivity study using fresh frozen human tissue arrays. This study did not identify any off-target binding for NI301A at 30 μg/mL, the highest concentration tested. No staining was observed on liver tissue sections, the main organ producing TTR, further confirming NI301A absent binding to physiological TTR protein (Supplementary Fig. 4).

**NI301A removes ATTR fibrils by macrophage-mediated phagocytosis.** By using human macrophages freshly prepared from peripheral blood mononuclear cells, we established NI301A's biological activity to trigger phagocytosis of fluorescent TTR-L55P aggregates. We used TTR-L55P because of its ability to oligomerize under physiological conditions. NI301A triggered ATTR-L55P phagocytosis in a concentration-dependent manner, with activity starting at low nanomolar antibody concentrations. Antibody-mediated phagocytosis of ATTR-L55P oligomers was impeded by blocking Fc receptors consistent with Fc-mediated phagocytosis. An isotype control antibody failed to trigger phagocytosis of ATTR-L55P (Fig. 4a). Flow cytometry experiments with ATTR-L55P and antibodies labeled with different fluorescent dyes showed that NI301A, but not the isotype control antibody, dose-dependently triggered the phagocytosis of ATTR-L55P oligomers by macrophages (Fig. 4b and Supplementary Fig. 5). Confocal microscopy of human macrophages during phagocytosis revealed intracellular vesicular localizations of ATTR-L55P and NI301A, suggesting co-localization of ATTR-L55P and NI301A within vesicles (Fig. 4b₃).

Amyloid deposits in patients consist of multiple proteins, proteoglycans, and the plasma protein Serum Amyloid P (SAP), potentially masking ATTR epitopes. To verify that NI301A triggers phagocytosis of patient amyloid deposits, we developed an ex vivo functional assay with fresh frozen myocardium sections containing ATTR and incubated them with human macrophages in the presence of antibodies. During 14 days of incubation, NI301A removed ATTR from human myocardium in a concentration-dependent manner. NI301A reduced both the total number of ATTR deposits and the total area covered by ATTR deposits by up to 50% (Fig. 4c, d). ATTR removal from patient myocardium tissue sections was specific for NI301A; the isotype control antibody failed to remove ATTR.

**NI301A-mediated complement activation accelerates amyloid clearance.** To further characterize NI301A's amyloid-removal activity, we evaluated the impact of complement activation on amyloid clearance. Phagocytosis assays were done in the presence or the absence of different concentrations of either fresh or heat-inactivated human plasma. Human plasma triggered a concentration-dependent increase in ATTR-L55P phagocytosis in vitro; heat inactivation of plasma abolished this activity (Fig. 5a). The acceleration by fresh plasma of ATTR phagocytosis was confirmed in ex vivo assays, resulting in an almost complete removal of cardiac amyloid deposits by macrophages (Fig. 5b). In vitro experiments further confirmed complement activation: NI301A binding to coated ATTRwt fibrils in the presence of fresh human plasma triggered formation of the insoluble complement factor C5b9, indicating that the complement cascade had been activated to completion (Fig. 5c). No complement activation occurred when NI301A was incubated with BSA and plasma, consistent with a specific effect of NI301A decorating ATTR fibrils.

**NI301A binds ATTR fibrils in vivo and activates amyloid clearance in mice.** To characterize the biological activity of NI301A in vivo, we developed a patient-derived amyloid xenograft (PDAX) mouse model. In PDAX mice, ATTR fibrils extracted from patient myocardium tissues were subcutaneously grafted and fluorescent NI301A-VT680 conjugates were injected intravenously (i.v.), to determine the target engagement of human amyloid in living mice (Fig. 6a). Human ATTR fibril grafts were thioflavin S-positive, indicating preserved β-sheet conformation of the ATTR amyloid deposits. Forty-eight hours after grafting and i.v. antibody administration, fluorescent NI301A-VT680 dose-dependently accumulated on human ATTR fibril grafts and colocalized with thioflavin S, indicating that it bound to human amyloid in vivo (Fig. 6b). The NI301A-VT680 fluorescence intensity increased linearly with dose over the range of 0.05–15 mg/kg, indicating that epitopes for antibody binding were not saturated with doses up to 15 mg/kg body weight (Fig. 6c).

For in vivo efficacy experiments, we produced the monoclonal mouse chimeric antibody ch.NI301A by fusing the human NI301A variable domains with murine IgG2a constant domain sequences. Six hours and 96 h following fibril grafting and injection of ch.NI301A, we measured ATTR by IHC. After 6 h of antibody treatment, mice presented similar amounts of ATTR per graft, corresponding to an average 86% of the graft area (range 83.2–89.7%). At this early time point, the fibril grafts were infiltrated by neutrophils (CD11B+, LY6G+, IBA1−), an

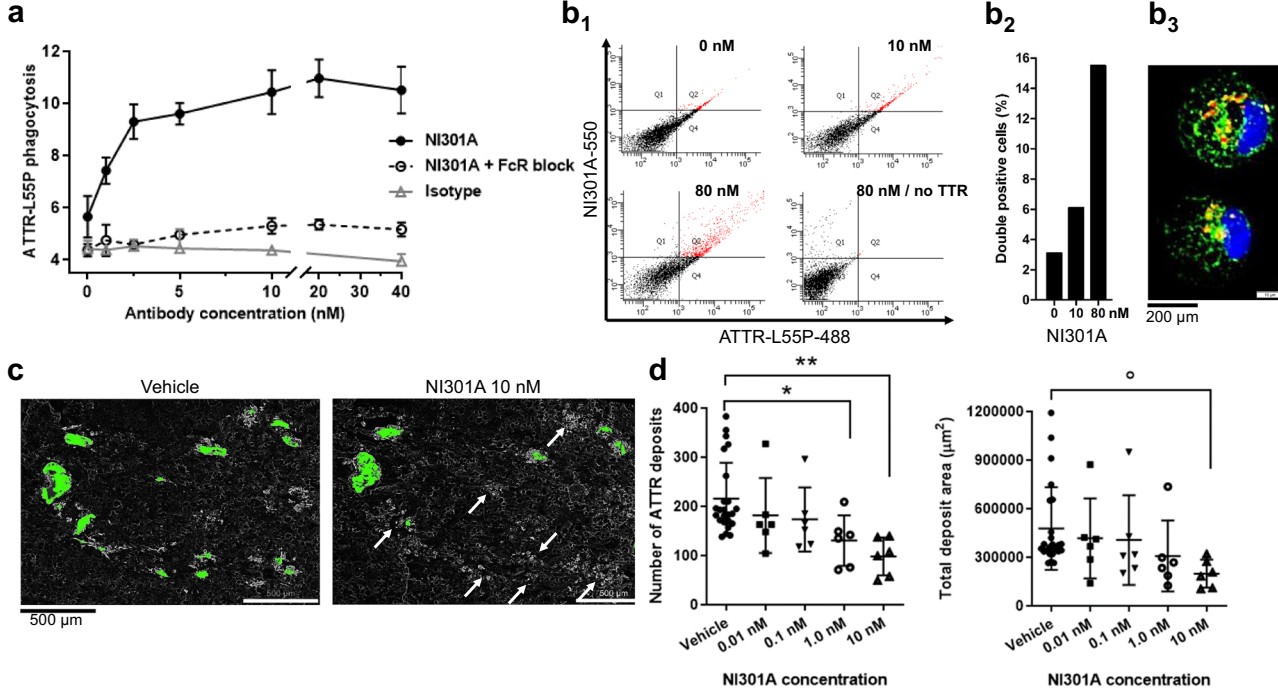

**Fig. 4 NI301A activates ATTR phagocytosis by human macrophages on human heart tissue. a** In vitro ATTR phagocytosis assay performed with human-derived macrophages, fluorescently labeled TTR-L55P protein, and different concentrations of NI301A or isotype control antibodies, with or without Fc-receptor blocking reagent (FcR block) (mean ± SD of triplicates). **b** In vitro ATTR phagocytosis with both TTR-L55P and NI301A fluorescent conjugates, and quantification of double-positive macrophages by FACS. **b₁** Dot plots; **b₂** quantification of double-positive cells; **b₃** representative double-positive macrophage imaged by confocal microscopy: blue: Dapi, green: TTR-L55P-atto488, red: NI301A-atto550. **c** Ex vivo tissue amyloid-removal assay performed with human-derived macrophages and NI301A on patient myocardium sections. Representative pair of adjacent tissue sections treated with NI301A or vehicle containing thioflavin S-positive amyloid deposits (quantified deposits in green). White arrows point to amyloid deposits removed by macrophages in the presence of NI301A. **d** Quantification of thioflavin S-positive amyloid deposit number and total area per tissue section following treatment with increasing concentrations of NI301A or control. Vehicle $n = 24$ sections, NI301A $n = 6$ sections per concentration (mean ± SD). One-way ANOVA and Dunnett's multiple comparisons test (mean difference and 95% CI). Number of deposits: $F_{(4, 43)} = 4.73$, $P = 0.003$; 1.0 nM vs. Vehicle: $-85.08$ ($-164.4$ to $-5.8$), *$P = 0.031$; 10 nM vs. Vehicle: $-117.4$ ($-196.7$ to $-38.1$), **$P = 0.0017$. Total deposit area: $F_{(4, 43)} = 1.91$, $P = 0.13$; 10 nM vs. Vehicle: $-278,662$ ($-560,983$ to $3660$), °$P = 0.054$.

immediate response characteristic of wound-healing process (Supplementary Fig. 6). After 96 h, the isotype-treated control group showed a moderate, but statistically significant, decrease in graft ATTR area, reflecting a spontaneous reaction against the human ATTR fibril grafts. All three doses (0.5, 5, and 50 mg/kg body weight) of ch.NI301A further reduced ATTR in a dose-dependent manner. Both the 5 and 50 mg/kg doses removed the amyloid fibrils almost completely, down to background levels of about 7% of the graft area. Even the low dose of 0.05 mg/kg ch. NI301A trended to remove amyloid, but its effect was not statistically significant (Fig. 6d). At this late time point (96 h), ATTR fibril grafts were entirely infiltrated by phagocytic monocytes (CD11B+, LY6G−, IBA1−) (Supplementary Fig. 6). Mice did not exhibit any sign of overt toxicity.

Collectively, the linear increase in target binding over the range of 0.05–15 mg/kg and the strong biological activity starting at doses of 5 mg/kg body weight suggested that target saturation was not required for the maximum activation of fibril removal by phagocytic immune cells in PDAX mice.

A similar in vivo experiment was performed using the Fc-inactive variant ch.NI301A-LALAPG at 10 mg/kg i.v. In contrast to treatment with ch.NI301A, which triggered ATTR fibril removal, treatment with ch.NI301A-LALAPG had no effect on fibril removal in vivo, indicating that effector functions are required for antibody-mediated amyloid clearance (Fig. 6e).

The effect of ch.NI301A was also evaluated at 10 mg/kg i.v. in the presence of the TTR stabilizer tafamidis, administered daily at

1 mg/kg intraperitoneally (i.p.). ch.NI301A fibril removal activity was maintained in the presence of tafamidis; tafamidis alone had no effect on ATTR fibril removal (Fig. 6f).

**Frequency of cardiac ATTR in autopsy samples from patients with heart failure.** To investigate the frequency of ATTR cardiomyopathy and further test the capacity of NI301A to bind ATTR in patients, we analyzed 80 autopsy heart tissue samples from patients who died of heart failure at ages ≥80 years. Varying amounts of Congo red and TTR antibody 39-44 IHC-positive ATTR deposits were present in 24 (30%) cases. The abundance of ATTR deposits was low in 11 samples (14%), moderate in 4 samples (5%), and high in 9 samples (11%) (Supplementary Table 1 and Supplementary Fig. 7). NI301A detected ATTR deposits by IHC in all 24 ATTR-positive samples.

**Discussion**
The results of this study show that NI301A, a human-derived amyloid-selective antibody targeting both ATTRwt and ATTRv, removed pathological ATTR deposits from patient myocardium tissues ex vivo, as well as patient-derived ATTR fibril grafts in mice in vivo. Characterization of the binding properties of NI301A demonstrated (1) that NI301A bound ATTR oligomers and fibrils with sub-nanomolar $EC_{50}$, (2) that NI301A was highly selective for the amyloid conformation of TTR, and (3) that NI301A similarly bound both wild-type and variant ATTR

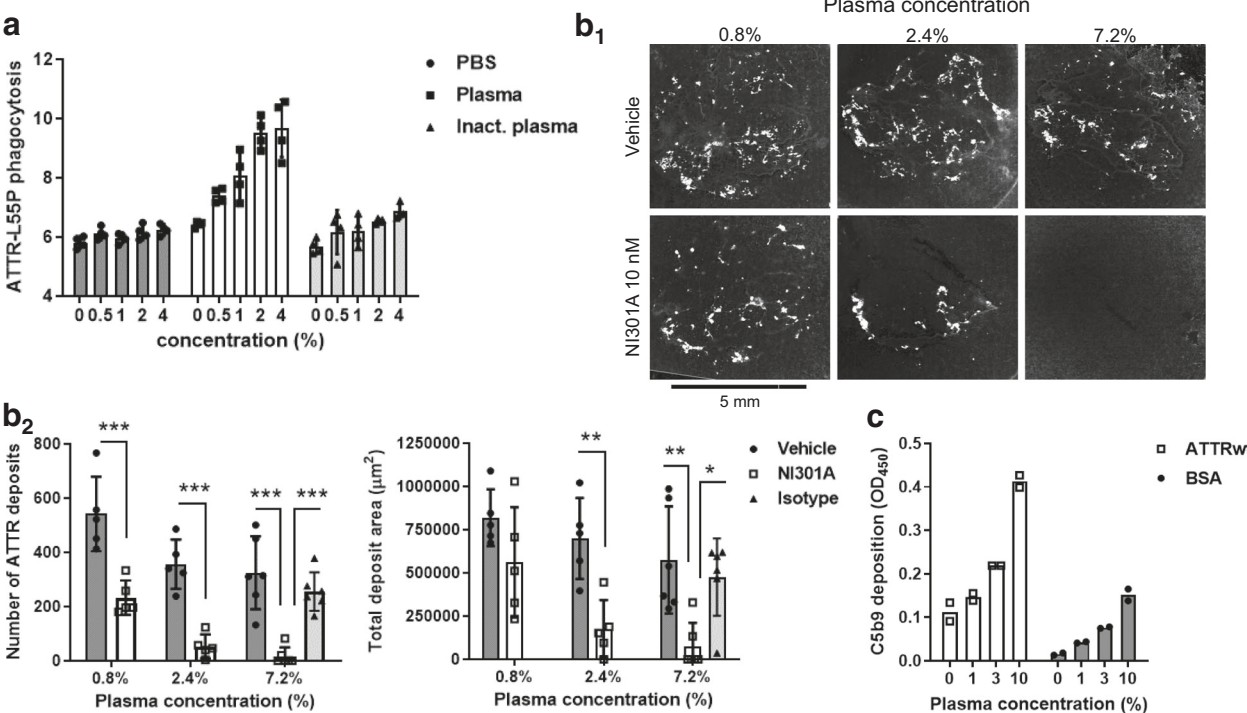

**Fig. 5 Antibody-mediated complement activation increases ATTR phagocytosis and tissue amyloid clearance. a** In vitro ATTR phagocytosis assay performed with human-derived macrophages, TTR-L55P-atto488, and NI301A in the presence of fresh or heat-inactivated human plasma at different concentrations (mean ± SD of quadruplicates). **b** Ex vivo tissue amyloid-removal assay performed with three concentrations of fresh human plasma. **b₁** Representative pairs of adjacent tissue sections treated with NI301A or vehicle. **b₂** Quantification of number and area of amyloid deposits per tissue section. $n = 5–6$ sections per condition (mean ± SD). Two-way ANOVA for main factors Plasma concentration and Antibody, and Bonferroni's multiple comparisons test of NI301A to vehicle conditions (mean difference and 95% CI). For the number of ATTR deposits: Plasma concentration: $F(2,26) = 16.04$, $P < 0.0001$; Antibody: $F(1,26) = 84.53$, $P < 0.0001$; NI301A vs. Vehicle using 0.8% plasma: −309.4 (−461.5 to −157.3), ***$P < 0.0001$; NI301A vs. Vehicle using 2.4% plasma: −302.6 (−454.7 to −150.5), ***$P < 0.0001$; NI301A vs. Vehicle using 7.2% plasma: −308 (−446.9 to −169.1), ***$P < 0.0001$; NI301A vs. Isotype using 7.2% plasma: −238.3 (−367.7 to −109.0), ***$P = 0.0007$. For total deposit area: plasma concentration: $F(2,26) = 6.85$, $P = 0.0041$; Antibody: $F(1,26) = 26.56$, $P < 0.0001$; NI301A vs. Vehicle using 2.4% plasma: −522,593 (−899,939 to −145,246), **$P = 0.0046$; NI301A vs. Vehicle using 7.2% plasma: −497,895 (−842,364 to −153,427), **$P = 0.0031$; NI301A vs. Isotype using 7.2% plasma: −398,001 (−734,405 to −61,596), *$P = 0.02$. **c** In vitro complement activation assay of NI301A in fresh human plasma using coated ATTRwt fibrils or bovine serum albumine (BSA) as target, followed by immunodetection of the complement activation product C5b9 (mean of duplicates).

(Fig. 1). The latter was confirmed by IHC using tissues collected from patients with sporadic or hereditary ATTR amyloidosis with cardiomyopathy or polyneuropathy (Fig. 2).

Selective binding to wild-type and variant ATTR oligomers and fibrils is consistent with results of structural studies showing that pathogenic mutations in the TTR polypeptide sequence can affect the stability of native tetramers, thus promoting their dissociation into monomers. These mutations, however, do not modify the propensity of TTR to misfold, oligomerize, and aggregate into amyloid-forming fibrils[14], suggesting that the pathogenic mechanisms of accelerated amyloid formation in the genetic forms of the disease is related to decreased stability of the native TTR tetramer.

We confirmed the conformation selectivity of NI301A with independent assays including sandwich ELISA, BLI and SPR, dot blot, semi-native PAGE and western blottings, IHC, and immunoprecipitation from human plasma (Figs. 1 and 2). This stringent selectivity for pathological TTR conformations results from NI301A binding a hidden linear epitope, which is not accessible on the native TTR conformation but becomes accessible upon misfolding of β-pleated sheets into a loop protruding from the amyloid core[3,15]. Conformation selectivity can be further enhanced by high avidity binding to repetitive epitopes that are displayed in oligomers and amyloid fibrils as was described for the β-amyloid-selective antibody aducanumab, which was

generated using a similar approach[16]. From a clinical perspective, the absent binding to physiological TTR tetramers and monomers may be a key requirement for therapeutic efficacy of NI301A, as off-target binding to native TTR present at high concentration in plasma could lead to impaired pharmacokinetics and reduced target engagement with tissue ATTR.

In human cardiomyopathy patients, large amounts of ATTR deposits accumulate in the myocardium. By using in vitro and ex vivo systems, we established that NI301A induces ATTR phagocytosis in an antibody concentration-dependent manner that is mediated by engagement of Fcγ receptors on human macrophages (Fig. 4). On heart tissue sections from human cardiomyopathy patients, NI301A dose-dependently removed tissue amyloid by triggering phagocytosis by human macrophages, attaining more than 50% amyloid removal within a 2-week time. This effect size is substantial, considering the absence of replenishment of macrophages in our ex vivo experiments (Fig. 4). Low concentrations of fresh human plasma further increased antibody-mediated amyloid removal from patient tissues, suggesting synergistic effects through complement factors in plasma (Fig. 5). The complement system can contribute to the clearance of amyloid in vivo by a combination of direct protease activity and *trans* signaling, triggering the formation of macrophage-derived multinucleated giant cells with the capacity to engulf very large particles[17]. Consistent with intact complement activity,

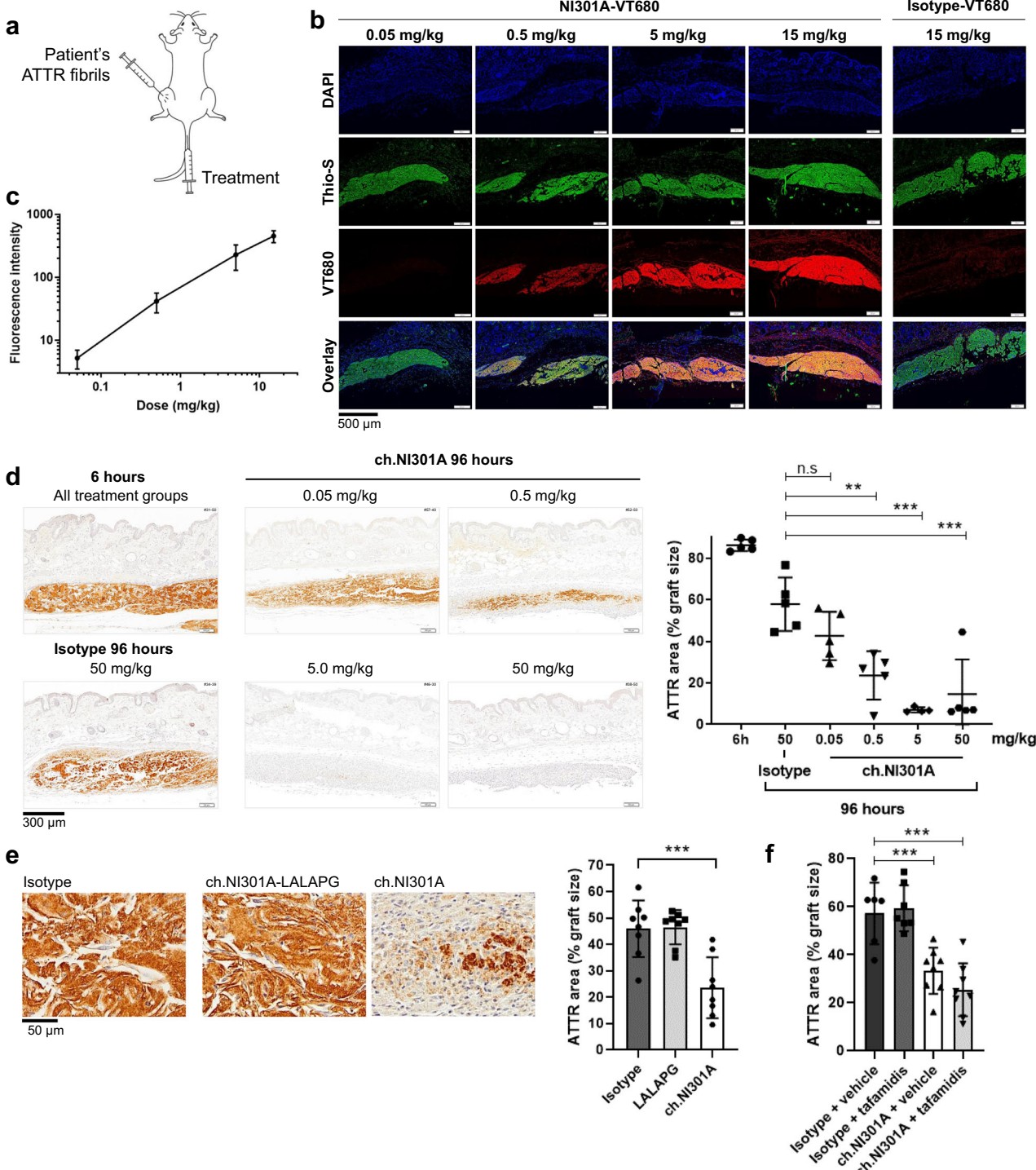

fresh, but not heat-inactivated, plasma increased NI301A-mediated phagocytosis of ATTR by macrophages (Fig. 5).

Finally, NI301A was active in vivo. It accumulated dose-dependently on human ATTR fibrils grafted into mice. A murinized chimeric variant of NI301A substantially accelerated the removal of ATTR by phagocytic cells and did not exhibit any signs of overt toxicity in this model. In good agreement with the in vitro phagocytosis data, antibody-mediated fibril removal in vivo was dose-dependent and required an active Fc domain (Fig. 6). Furthermore, the antibody-mediated fibril removal activity was maintained in the presence of tafamidis, currently the only approved therapy for ATTR-CM.

ATTR cardiomyopathy is characterized by large accumulation of extracellular ATTR fibrils in the myocardium, which can lead to a doubling of extracellular volumes in patients as compared to healthy subjects[18,19], an increase of the left ventricle wall thickness from <12 mm in healthy subjects to >20 mm in advanced patients[20,21], and ATTR deposition occupying 30–40% of the total myocardium tissue area[22]. Mechanically, amyloid fibrils are remarkably rigid and their resistance to rupture has been compared to that of steel[23]. As a result, the accumulation of ATTR in the myocardium causes progressive tissue rigidification together with loss of muscle elasticity. Functionally, this loss in elasticity can cause diastolic dysfunction during initial disease stages

**Fig. 6 Dose-dependent binding of NI301A on TTR amyloid in vivo and activation of immune clearance in mice. a** Schematic of the patient-derived amyloid xenograft mouse model: ATTR fibril graft was placed subcutaneously on the thigh and treatments administered i.v. **b** In vivo binding of fluorescently labeled NI301A-VT680 to patient-derived ATTR fibril grafts 48 h after amyloid grafting and antibody administration. **c** Quantification of NI301A-VT680 fluorescence intensity on ATTR grafts (mean ± SD of 5 mice per dose level using 3 sections per mouse). **d** ch.NI301A-mediated clearance of ATTR grafts in vivo. ATTR remaining after 6 and 96 h of treatment was detected by IHC staining and quantified (mean ± SD of $n = 4$–5 mice per group using 3–6 sections per mouse). One-way ANOVA ($F_{(4, 19)} = 13.6$, $P < 0.001$) and Dunnett's multiple comparisons test (mean difference and 95% CI). Ch. NI301A 0.5 vs. Isotype: $-34.3$ ($-55.0$ to $-13.6$), **$P = 0.001$; ch.NI301A 5 vs. Isotype: $-50.9$ ($-72.8$ to $-28.9$), ***$P < 0.001$; ch.NI301A 50 vs. Isotype: $-43.4$ ($-64.1$ to $-22.6$), ***$P < 0.001$. **e** Evaluation of Fc-silent ch.NI301A-LALAPG antibody on ATTR graft removal in vivo (mean ± SD of $n = 8$ mice per group using 3–4 sections per mouse, all antibodies at 10 mg/kg i.v.). One-way ANOVA ($F_{(2, 21)} = 14.2$, $P = 0.0001$) and Dunnett's multiple comparisons test. Ch.NI301A vs. isotype: $-22.4$ ($-34.0$ to $-10.7$), ***$P = 0.0003$. **f** Evaluation of ch.NI301A + tafamidis combination on ATTR removal in vivo (mean ± SD of $n = 6$–8 mice per group, 3–4 sections per mouse; antibodies once at 10 mg/kg i.v. and tafamidis daily at 1 mg/kg i.p.). Two-way ANOVA for main factors Antibodies and Stabilizer, and Dunnett's multiple comparisons test to the isotype + vehicle group. Antibody: $F_{(1, 25)} = 52.3$, $P < 0.0001$; Stabilizer: $F_{(1, 25)} = 0.52$, $P = 0.48$. Ch.NI301A + vehicle vs. isotype + vehicle: $-23.9$ ($-38.2$ to $-9.5$), ***$P = 0.001$; ch.NI301A + tafamidis vs. isotype + vehicle: $-31.8$ ($-46.1$ to $-17.4$), ***$P < 0.0001$.

followed by reductions in ejection fraction and heart failure. Both the reduction in left ventricle ejection fraction and the loss of contractility are strongly correlated with disease severity and reduced survival time[20,24]. Several therapeutic approaches for the treatment of ATTR amyloidosis are focused on a slowing or the prevention of amyloid formation. The TTR stabilizers tafamidis and the repurposed anti-inflammatory drug diflunisal, as well as TTR gene silencers patisiran and inotersen, have achieved substantial reductions or even a complete halt in disease progression[25–28]. NI301A has a different mode-of-action that is targeting the active removal of pre-existing amyloid and may therefore have a potential beyond disease stabilization for functional improvements by decreasing cardiac muscle stiffness and improving heart contractility and elasticity.

We evaluated the frequency of ATTR cardiomyopathy in elderly male Caucasian patients who died of heart failure with preserved ejection fraction and found pathologic amounts of ATTR in 16% of the samples. This high frequency is consistent with published data, indicating that ATTR cardiomyopathy is a massively underdiagnosed condition[6–11]. NI301A bound ATTR deposits in all positive tissues, suggesting a broad applicability across patients.

Antibody-mediated amyloid removal is a novel treatment modality. It is currently tested in advanced stages of clinical development with proof-of-mechanism obtained in patients. In Alzheimer's disease, the Aβ-targeting antibodies aducanumab and gantenerumab achieved substantial reductions in Aβ plaque load in patients[29,30]. Amyloid removal increased with dose and time. It was associated with amyloid-related imaging abnormalities, potentially a correlate of the biological mechanism of antibody-mediated amyloid removal[31]. Various other antibodies targeting amyloid or pathological protein aggregates are under clinical investigation. These include the α-synuclein aggregate-selective antibodies cinpanemab and PRX002/RG7935 for the potential treatment of Parkinson's disease[32], the anti-τ antibody BIIB076 for the potential treatment of Alzheimer's disease[33], and the misfolded SOD1-selective antibody AP101 for the potential treatment of amyotrophic lateral sclerosis[34]. Dezamizumab is a humanized monoclonal IgG1 antibody binding to native SAP component, a plasma protein identified in all types of amyloid deposits. In a phase 1 clinical trial conducted in patients with systemic amyloidosis, dezamizumab treatment was associated with signs of organ response such as decreases in liver stiffness, which were highly consistent with the macrophage-dependent phagocytic clearance of amyloid demonstrated in animal models[17,38,39]. A phase II trial in patients with cardiac amyloidosis was initiated but terminated prematurely due to an apparent change in the risk-benefit profile, which might have been related to dezamizumab binding physiological SAP protein.

Similar to aducanumab, cinpanemab, and AP101, NI301A is an aggregate-selective antibody that was identified based on immune repertoire analyses of memory B-cell complements from healthy elderly subjects. Such antibodies targeting pathogenic conformations of self-proteins may be generated in response to the age-associated accumulation of pathological protein aggregates including various form of amyloid, generating the possibility of a physiological mechanism, for elimination of misfolded proteins by engaging immune biology otherwise targeting non-self, foreign proteins. As a corollary, antibody-mediated removal of ATTR amyloid from heart tissues may well represent an additional approach for the treatment of patients with cardiomyopathy caused by ATTR amyloidosis. To test this possibility, NI301A (NI006) is currently evaluated in a phase 1 clinical trial in ATTR-CM patients.

## Methods

**Antibody identification, cloning, and recombinant expression.** cDNA sequences encoding human ATTR-binding antibodies were derived from a de-identified blood lymphocyte library collected from healthy elderly subjects. Memory B cells were isolated and cultivated as described in Sevigny at al.[29]. Human memory B cells were CD22+, CD27+ and IgD−, IgM−, CD3−, CD56−, and CD8a−. They were screened by ELISA for the expression of antibodies binding to wild-type and variant ATTR. Positive hits were counter-screened to exclude clones cross-reacting with unrelated amyloid-forming proteins. Selected ATTR-reactive B-cell clones were subjected to cDNA cloning of IgG heavy and light-chain variable region sequences, and sub-cloned into expression vectors encoding human IgG1 constant domain sequences. Expression constructs encoding IgG1 heavy and light chains were transiently expressed in CHO-S cell line and purified by protein A affinity chromatography.

Human blood samples from healthy elderly subjects were collected under written informed consent and study approval by the Ethics Committee of the Canton of Zurich. The study was compliant with the Helsinki Declaration.

**Preparation of human ATTR.** Human TTRwt was obtained by purification from human blood plasma (Bio-Rad). The pathogenic TTR protein variants V20I, V30M, L55P, S77Y, S112I, Y116S, T119M, and V122I were expressed recombinantly in *Escherichia coli* (Alexotech and Wako); pure preparations of non-amyloid, native TTRwt, and TTRv tetramers were obtained by size-exclusion chromatography on Superdex 75 columns, to remove pre-existing misfolded proteins. ATTR fibrils were prepared by aggregating TTR at 200 µg/mL in 50 mM acetate-HCl, 100 mM KCl, 1 mM EDTA pH 3.0 for 5 days at 65 °C under shaking conditions at 1000 r.p.m.[15]. This procedure generated thioflavin S-positive fibrils readily detectable by electron microscopy (Fig. 1a). Water-soluble ATTR oligomers were prepared with the above protocol and by limiting the aggregation time of TTR to 4 h at 37 °C.

**Enzyme-linked immunosorbent assay.** ELISA assays were done in 96-well microplates (Corning) coated with ATTR fibrils and were blocked with 2% BSA, 0.1% Tween-20 in phosphate-buffered saline (PBS) before adding primary antibodies in duplicates for overnight incubation at 4 °C. After washing, bound antibodies were detected with horseradish peroxidase (HRP)-conjugated anti-human IgG antibodies (Jackson Immunoresearch) and DAB (3,3'Diaminobenzidine) substrate. The absorbance was measured using a multifunction plate reader (ThermoFisher Varioskan Lux and SkanIt 6.02 software). Data were exported to

Excel (Microsoft) for data management and were fitted with a logarithmic dose–response function with variable slope (four parameter logistic regression, 4PL) using the least-square regression in Prism 7.03 (GraphPad).

**Capture ELISA.** Streptavidin-coated plates (Greiner) were loaded with biotinylated NI301A and blocked before the addition of ATTR or TTR samples in duplicates for 2 h incubation at room temperature. After washing, bound ATTR was detected with an HRP-conjugated NI301A antibody in combination with luminescent substrate (Biotinylation kit, HRP-labeling kit, and HRP substrate from Thermo-Fisher). Data were fitted using 4PL with $1/y^2$ weighting.

**Binding kinetics assays.** Binding to soluble ATTRwt oligomers was determined by SPR on a ProteOn XPR36 machine (Bio-Rad) at 25 °C in HBS-T buffer, using an anti-human IgG, Fcγ-specific antibody (Jackson Immunoresearch) covalently coupled to a GLM sensor chip, to capture NI301A on the sensor. Data were analyzed in ProteOn Manager v. 2.0 and double-referenced using interspots and the buffer-only channel, and fitted by using a 1 : 1 kinetic model (Langmuir fit).

Binding to soluble ATTRv oligomers was determined by BLI on an Octet RED96 machine (Pall ForteBio). As this technology circumvents the need for microfluidics, it is well suited for measuring binding to protein aggregates. The experiments were done at 25 °C by using streptavidin sensors preloaded with biotinylated NI301A or isotype control antibodies. Data were analyzed in DataAnalysis v.8.2 and double-referenced using isotype and buffer-only conditions, and fitted using a 1 : 1 kinetic model.

**Immunoblottings.** For dot blots, 200 ng of TTR tetramers and ATTR fibrils prepared as described above were immobilized on nitrocellulose membranes by using vacuum filtration. For western blottings, TTR-V30M was aggregated for 4 days at 37 °C under near-physiological conditions in 10 mM citric acid pH 6.0, 150 mM NaCl, and 5 mM Ca2+. Samples were separated by non-reducing SDS-PAGE with no heat denaturation, to preserve the aggregates' sizes, and were immobilized on nitrocellulose membranes by semi-dry blotting (ThermoFisher). Epitope mapping was conducted using a cellulose membrane array of 29 overlapping peptides of 15 amino-acid length and 11 amino-acid overlap, which covered the whole mature TTR protein sequence, and 15 additional peptides which covered the TTR leader peptide sequence and 13 selected TTR mutations associated with hereditary TTR amyloidosis. Membranes were blocked, incubated with NI301A overnight at 4 °C, and developed with HRP-conjugated, anti-human, rabbit or mouse IgG antibodies (Jackson Immunoresearch) in combination with a chemiluminescent substrate (ThermoFisher).

**Immunoprecipitation.** Protein A-coated magnetic beads (Dynabeads, Life Technologies) were saturated with NI301A or anti-TTR Dako A0002 antibodies and incubated for 30 min at room temperature with TTRwt tetramers, or with patient-derived ATTR fibrils functioning as controls, or with human plasma samples diluted 0, 10, 100, 1000, and 10,000 times in PBS with Tween-20. After washing, the antibody–antigen complexes were eluted directly in SDS-PAGE loading buffer (80 °C for 15 min) separated by SDS-PAGE and analyzed by western blotting with the Dako A0002 TTR antibody.

**Immunostaining.** Frozen or paraformaldehyde (PFA)-fixed tissue biopsies obtained from ATTR-CM patients and from ATTR polyneuropathy patients were stained with Congo red by using Putchler's modifications[35]. Immunostainings were done after quenching endogenous peroxidase activity with 3% $H_2O_2$ in methanol, blocking with 4% BSA and 5% normal horse and goat sera in PBS, incubation with primary antibodies overnight at 4 °C, and detection using goat anti-rabbit IgG antibody (Jackson Immunoresearch) where needed, in combination with the Vectastain ABC kit (Vector Laboratories) and diaminobenzidine (Dako). NI301A was used as a biotinylated conjugate at 10 nM on frozen tissue sections. Antibody Dako A0002 was used at 1 : 500 or 1 : 1000 dilutions and required PFA post fixation for staining, leading to a different appearance of the tissues due to the different processing. Tissue scanning microscopy was done at ×20 magnification in bright field and polarization modes with an Olympus VS120 virtual slide microscope.

Immunostaining of PFA-fixed paraffin-embedded ATTR fibril grafts in mice was performed as described above using the following primary antibodies: Dako A0002 for TTR, Abcam ab133357 for CD11B, Abcam ab2557 (clone NIMP-R14) for LY6G, and Wako 019-19741 for IBA1.

Tissue cross-reactivity analyses were conducted by immunostaining using a biotin-conjugated NI301A antibody at 1, 10, and 30 µg/mL on frozen human tissue arrays (USBiomax FDA903). The arrays contained 90 tissue sections representing 30 different organs each from three different individuals.

**Fibril extraction from tissues.** ATTR fibrils were extracted from frozen cardiac tissues obtained at autopsy and semi-purified using the detergent-free protocol described by Pras et al.[36]. This protocol preserves the amyloid conformation by relying on the partial solubility of amyloid fibrils in pure water.

**In vitro and ex vivo phagocytosis assays.** Macrophages were differentiated from fresh human monocytes by culturing these for 10–15 days in macrophage serum-free medium (M-SFM, Life Technologies) supplemented with macrophage colony-stimulating factor (100 ng/mL M-CSF, Miltenyi), or granulocyte M-CSF (50 ng/mL GM-CSF, Sigma).

In vitro phagocytosis was done for 2 h at 37 °C in the presence of fucoidan at 0.5 mg/mL, ATTR-L55P labeled with Atto488 (Sigma) at 7 µg/mL, NI301A, and isotype control at concentrations from 0 to 80 nM. An Fc-receptor inhibitor (Miltenyi) was used for negative control at 1 : 100 dilution. After washes, the levels of ATTR-L55P-atto488 internalization were measured with a fluorescence plate reader (Varioskan Lux, ThermoFischer). Antibodies labeled with Atto550 (Sigma) were also used in combination with fluorescence-activated cell sorting to quantify double-positive macrophages only. All experiments were performed in triplicates. For confocal microscopy imaging, experiments were done with macrophages coated on coverslips.

Ex vivo phagocytosis was done with human macrophages on 15 µm-thick post-mortem myocardial sections obtained from ATTR-CM patients. Tissue sections were mounted on glass coverslips, using pairs of adjacent sections for antibody or vehicle control conditions. The M-SFM/GM-CSF medium was supplemented with interferon-γ (100 ng/ml) and lipopolysaccharide (20 ng/ml) to stimulate macrophage phagocytic activity. Tissue sections (6 replicates per condition) were incubated for 14 days at 37 °C (95% $CO_2$, 5% $O_2$) with 50% replacement of the cell culture medium every 3–4 days; they were then directly fixed with 4% PFA, washed, stained with thioflavin S, and mounted. The full area of the tissue sections was scanned using an automated slide scanner (Olympus VS120), at ×20 magnification in both 4′,6-diamidino-2-phenylindole and tetramethylrhodamine channels. Both numbers of and areas covered by ATTR deposits were quantified by using an automated image analysis algorithm implemented in a commercial image analysis software (Image-Pro Premier, Media Cybernetics). Data were analyzed by one-way analysis of variance (ANOVA) followed by Dunnett's multiple comparison tests.

For all in vitro and ex vivo experiments with human plasma, plasma was used either freshly or inactivated by heating at 56 °C for 30 min.

**Complement activation assay.** ELISA plates coated with ATTRwt fibrils or BSA were loaded with NI301A and incubated with increasing concentrations of fresh human plasma for 1 h at 37 °C. After washing, the complement complex C5b9 was detected with a specific antibody (Abcam) and corresponding secondary antibody (Jackson Immunoresearch).

**PDAX model.** Animal experiments were conducted in compliance with the Swiss federal regulations and received approval from the Swiss Veterinary Service (license ZH269-14). ATTR fibrils were obtained from post-mortem heart tissues of ATTR-CM patients using the detergent-free water-based extraction procedure[36] and were prepared at 2 mg/mL concentration to confer high viscosity to the extract. ATTR fibrils were grafted into SKH1-Elite female mice of 2.5–3 months of age by subcutaneous injection of 50 µL in the thighs followed by i.v. administration of the antibody. NI301A antibody was labeled with the fluorescent dye VivoTag 680 (Perkin Elmer) for target binding experiments (NI301A-VT680) and the mouse chimeric IgG2a NI301A variant (ch.NI301A) was used for in vivo efficacy experiments. An Fc-inactive variant, named ch.NI301A-LALAPG, was generated using the L234A, L235A, and P329G substitutions, which are known to eliminate Fcγ receptor and complement factor C1 binding[37]. Tafamidis (Carbosynth) was prepared at 100 µg/mL in vehicle (PBS 7.4, 10% EtOH, 10% PS80) and administered daily at 1 mg/kg i.p. Data were analyzed by one-way or two-way ANOVAs followed by Dunnett's multiple comparisons test using the isotype-treated group as reference.

**Human heart tissue analysis.** The National Disease Research Interchange collected the autopsy heart tissues from males with heart failure, with preserved ejection fraction, who died at an age of 80 years or more. In all cases, the donors (or their next-of-kin) provided informed consent to procure biospecimens for biomedical research. Only donors on the Center for Disease Control High Risk category and donors with cardiac tumors were excluded from the collection. Samples were processed for histological analysis. Congo red-positive samples were stained by IHC with TTR antibodies Dako A0002, 39-44 (Alexotech) and with NI301A as described above.

**Statistics and reproducibility.** All group values are presented as mean and the SD represented for groups with $n > 3$. Statistical analysis were performed using the GraphPad Prism software, using one-way ANOVA and Dunnett's multiple comparisons test or two-way ANOVA and Bonferroni's multiple comparison test. $P < 0.05$ was considered as statistically significant. All experiments have been repeated on independent occasions and generated similar results.

**Reporting summary.** Further information on research design is available in the Nature Research Reporting Summary linked to this article.

## Data availability

All unique biological material included in this article can be made available upon substantiated request to the corresponding author. Source data are provided with this paper.

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

## Acknowledgements

We thank the patients and their relatives who donated the biospecimens used in this study, and acknowledge partial funding through a Eurostars grant (E! 9036). We also thank the National Disease Research Interchange for providing biospecimens; Dr. Fabrice Heitz, Neurimmune, and Gery Barmettler, Center for Microscopy and Image Analysis University of Zurich, for electron microscopy imaging; Dr. Stefan Schauer, Functional Genomics Center ETH and University of Zurich, for his support with kinetic binding experiments using SPR; and the Antibody Technology Group at Neurimmune for antibody cloning and expression.

## Author contributions

A.M. and J.G. conceived and led the study. A.M., A.H., C. Huy, E.V. and B.C. designed and/or performed, and/or analyzed experiments. C. Hock, T.D., O.B.S. and M.J.S. provided materials and scientific guidance. B.C., C. Hock, R.M.N. and J.G. contributed to experimental design and interpretation of data, and critically revised the manuscript. A.M. wrote the manuscript.

## Competing interests

A.M., A.H., C. Huy, E.V., B.C., C. Hock, R.M.N. and J.G. are or were (E.V.) employees of Neurimmune. O.B.S. participates in clinical trials for Alnylam and Prothena Pharmaceuticals, and has served in Advisory boards for Pfizer, Alnylam, Akcea, Prothena, and Intellia Pharmaceuticals, and has participated in educational activities sponsored by Alnylam and Akcea Pharmaceuticals. T.D. participates in clinical trials and has served in Advisory boards for Alnylam, Akcea, Pfizer, and Prothena Pharmaceuticals. M.J.S. does not have competing interest.
