## [Peer Review File · Nature Communications]

REVIEWER COMMENTS

Reviewer #1 (Remarks to the Author):

The tetramer of transthyretin (TTR) is the physiological form abundantly presented in circulation, wild type TTR (TTRwt) can dissociate into dimers and oligomers, which in turn misfold and aggregate leading to transthyretin amyloid (ATTR) deposition. Several genetic mutations in the TTR gene (TTRv) can destabilize the TTR tetramer, which significantly contributes to ATTR pathologies. Current therapeutic strategies for diseases related to ATTR pathologies aim to silence TTR production or to stabilize the TTR tetramer, however, treatment options aiming to reduce already formed ATTR pathologies are not available yet.

The authors present a novel IgG monoclonal antibody against ATTR (clone NI301A), which has been isolated from human memory B cells of healthy elderly subjects. The NI301A antibody recognized various pathological forms of ATTR, including ATTR oligomers, ATTR fibrils, and various sizes of ATTR aggregates. The binding affinity of NI301A to ATTRwt oligomers determined by surface plasmon resonance is considerable ($K_D = 1.2$ nM). Conversely, NI301A does not recognize the TTR tetramer. The authors argue that the NI301A binding epitope located at AA position 41 to 45 of TTR, which is hidden in the native monomer, is exposed under pathological conditions. In cultured patient myocardium tissue sections with ATTR pathologies *ex vivo*, NI301A antibody enhanced Fc-dependent macrophage phagocytosis and clearance of ATTR deposition. In a xenograft mouse model, NI301A can remove subcutaneously grafted human ATTR in a dose-dependent manner *in vivo*. NI301A seems a druggable candidate to reduce ATTR pathologies, and therefore, are of great interest to the field

Major concerns:

1. Phagocytosis by macrophages is critical for NI301A-mediated ATTR clearance *in vivo*, while, the inflammatory responses toward xenografts in mice are not characterized in current study. They might be quite different when compared to the inflammatory responses in cardiomyopathy. Such differences might be a concern when attempting to translate the current study into human clinic trials.
2. NI301A is isolated from memory B cells of healthy elderly subjects. The surface markers of these memory B cells should be listed. The authors argued that such antibody responses may be generated in response to age-associated accumulation of pathological proteins. It would be important to know the frequency of antibody responses toward the pathological forms of ATTRs in healthy elderly subjects versus patients with ATTR. Such data should be provided.
3. How many types of human tissue have been used to stain with the NI301A antibody? Tissue arrays using multiple tissues might be helpful to confirm and verify the selective staining pattern of NI301A for pathological forms of TTR, i.e., the tissues that highly expressed TTR, and the organs that are highly influenced by ATTR pathologies.

Minor concerns:

1. Is the NI006 antibody currently used in a ATTR cardiomyopathy clinic trial identical to the NI301A antibody?
2. The scale bar is missing in Figure 4 B3. The size of the scale bar is missing or not visible or provided in the remainder of images with IF/IHC staining.
3. The protein size in Figure 1H should be indicated or a size marker should be provided. Is this the only protein band after immunoprecipitation with human serum?
4. Please double check the bar size of Figure 2A, A - control of DAKO A0002 staining image. It appears to use a different magnification when compared to the others, but an identical scale bar is given.
5. How many replicate experiments have been performed in Figure 4B1 and B2?

Reviewer #2 (Remarks to the Author):

This study represents to the development of an antibody that is specific only for miss folded transthyretin and does not bind to transthyretin in the native tetra her state. The binding is specific to the Fc fragment of the immunoglobulin and is enhance by the use of complement.

The antibody binds to 5 amino acids that are not exposed in native transthyretin but only exposed when miss folded both in mutant as well as wild-type amyloid deposits. The controls used in this trial are appropriate in the antibody was tested in multiple different mutations of amyloid.

In vitro binding occurs with the antibody were commercial anti transthyretin anti serum fails to bind. The antibody can also buying to amyloid deposits placed in mice and triggers phagocytosis resulting in resolution of the deposits. 9/80 autopsy he has for heart failure patients over the age of 80 demonstrated ATTR deposits suggesting this is a widespread problem for which this antibody offers a potential solution beyond the currently available stabilizing therapies.

The only question I have is whether any toxicity studies were done on the mice that had fibro graft placement followed by injection of monoclonal mouse chimeric antibody.

Reviewer #3 (Remarks to the Author):

In their paper "A human antibody selective for transthyretin amyloid removes cardiac amyloid through phagocytic immune cells" Michalon et al present data on a new antibody therapy for Transthyretin amyloid cardiomyopathy, a rare condition which is characterised by the deposition of extracellular fibrils in the myocardium. Reducing myocardial fibrils load as a therapeutic goal has been demonstrated before which reduces the novelty of the work. The authors present NI301A, a monoclonal antibody obtained from memory B cell of healthy elderly subjects.

The paper is well written, the data backs up the claims made and the authors present a large amount of in vitro and in vivo data. They demonstrate that the antibody can dissolve fibrils in vitro (well controlled with heat inactivated plasma etc) and in vivo using an elegant patient derived xenografts model with a purpose made human/mouse chimeric antibody.

Overall, this is a strong and important study tackling a rare but life threatening disease.

My main comment is that whilst the authors discuss the use of other approaches such as small molecule therapeutics for the indication e.g. tafamidis and diflunisal as well as gene therapeutics patisiran and inotersen they ignore other therapeutics approaches such as Dezamizumab. To provide convincing data that the selected antibody approach is superior in reversing existing disease rather than stopping progression it is essential to do a head-to-head comparison in vitro but most importantly in vivo with competing antibodies and perhaps also other approach currently under investigation.

"A human antibody selective for transthyretin amyloid removes cardiac amyloid through phagocytic immune cells"

RESPONSE TO REFEREES

Reviewer #1 (Remarks to the Author):

The tetramer of transthyretin (TTR) is the physiological form abundantly presented in circulation, wild type TTR (TTRwt) can dissociate into dimers and oligomers, which in turn misfold and aggregate leading to transthyretin amyloid (ATTR) deposition. Several genetic mutations in the TTR gene (TTRv) can destabilize the TTR tetramer, which significantly contributes to ATTR pathologies. Current therapeutic strategies for diseases related to ATTR pathologies aim to silence TTR production or to stabilize the TTR tetramer, however, treatment options aiming to reduce already formed ATTR pathologies are not available yet.

The authors present a novel IgG monoclonal antibody against ATTR (clone NI301A), which has been isolated from human memory B cells of healthy elderly subjects. The NI301A antibody recognized various pathological forms of ATTR, including ATTR oligomers, ATTR fibrils, and various sizes of ATTR aggregates. The binding affinity of NI301A to ATTRwt oligomers determined by surface plasmon resonance is considerable ($K_D = 1.2$ nM). Conversely, NI301A does not recognize the TTR tetramer. The authors argue that the NI301A binding epitope located at AA position 41 to 45 of TTR, which is hidden in the native monomer, is exposed under pathological conditions. In cultured patient myocardium tissue sections with ATTR pathologies *ex vivo*, NI301A antibody enhanced Fc-dependent macrophage phagocytosis and clearance of ATTR deposition. In a xenograft mouse model, NI301A can remove subcutaneously grafted human ATTR in a dose-dependent manner *in vivo*. NI301A seems a druggable candidate to reduce ATTR pathologies, and therefore, are of great interest to the field.

Major concerns:

1. Phagocytosis by macrophages is critical for NI301A-mediated ATTR clearance *in vivo*, while the inflammatory responses toward xenografts in mice are not characterized in current study. They might be quite different when compared to the inflammatory responses in cardiomyopathy. Such differences might be a concern when attempting to translate the current study into human clinic trials.

Response:

To address the referee's comments, we have added an analysis of the inflammatory response towards xenografts in mice based on immunohistochemical staining for CD11B, a myeloid lineage marker, LY6G, a granulocyte marker, and IBA1, a macrophage marker, 6 hours and 4 days after fibril graft. We observed that the xenograft was rapidly infiltrated by neutrophils (CD11B+, LY6G+, IBA1-), consistent with a typical wound healing process. Four days after the fibril graft, neutrophils were replaced by phagocytic monocytes (CD11B+, LY6G-, IBA1-). ch.NI301A treatment substantially accelerated the removal of grafted amyloid fibrils (Fig. 6D-E).

We have added the new supplementary figure 5, complemented the material and method sections and added the following text to the manuscript:

“At this early time point, the fibril grafts were infiltrated by neutrophils (CD11B+, LY6G+, IBA1-), an immediate response characteristic of wound healing process (supplementary Fig. 5).”

“At this late time point (96 hours), ATTR fibril grafts were entirely infiltrated by phagocytic monocytes (CD11B+, LY6G-, IBA1-) (supplementary Fig. 5).”

These data, together with the *ex vivo* efficacy data demonstrating amyloid removal from patient myocardium tissue sections, provided the basis for investigating the therapeutic effects of NI006 treatment in patients.

We agree with the reviewer that the inflammatory responses in human cardiomyopathy patients might differ from the responses observed in the fibril xenograft model. The ongoing phase 1 clinical trial is assessing the safety of this therapeutic approach by a careful dose-escalation, extensive clinical monitoring, and frequent measurements of cytokine levels.

2. NI301A is isolated from memory B cells of healthy elderly subjects. The surface markers of these memory B cells should be listed. The authors argued that such antibody responses may be generated in response to age-associated accumulation of pathological proteins. It would be important to know the frequency of antibody responses toward the pathological forms of ATTRs in healthy elderly subjects versus patients with ATTR. Such data should be provided.

Response:

To address the referee’s first comment, the following text has been added to the Materials and Method section: *“Human memory B cells were CD22+, CD27+ and IgD-, IgM-, CD3-, CD56- and CD8a-.”*

We also determined the frequency of antibody responses toward the pathological forms of ATTRs in our de-identified library of healthy elderly subjects. We observed an overall frequency of ATTR-reactive memory B cells of about 1 in 55’000 memory B-cells. We hypothesize that of the abundantly expressed TTR protein, small amounts of misfolded species can form even during normal ageing and the corresponding neoepitopes may trigger immune responses in healthy-individuals including the formation of memory B-cells. These observations are consistent with our findings for other aggregation prone proteins or peptides such as Abeta, alpha-synuclein, or tau which have been shown to accumulate already decades before the manifestation of the first symptoms of disease. Using our technology of immune repertoire analysis in elderly healthy donors, we have identified corresponding B-cell memory for many of such aggregate forming proteins, suggesting that this is a common immune mechanism (c.f. Sevigny et al., 2016; Weihofen et al., 2019; Nobuhara et al., 2017; Maier et al, 2018). We agree with the reviewer that it could be of interest to determine the corresponding frequency of memory B-cells in ATTR patients. This would however require a new clinical study with a large sample collection to generate a sizeable library of memory B-cell inventories from TTR patients that would subsequently have to be screened in extensive high throughput screening campaigns.

3. How many types of human tissue have been used to stain with the NI301A antibody? Tissue arrays using multiple tissues might be helpful to confirm and verify the selective staining pattern of

NI301A for pathological forms of TTR, i.e., the tissues that highly expressed TRR, and the organs that are highly influenced by ATTR pathologies.

Response:

To address the reviewer's point, we have added the evaluation of NI301A binding selectivity using fresh frozen human tissues. This tissue cross-reactivity study did not identify any off-target binding for NI301A even at high concentrations.

We have added the new supplementary figure 4, complemented the material and method sections and added the following text to the manuscript:

"The selectivity of NI301A was further confirmed in a human tissue cross reactivity study using fresh frozen human tissue arrays. This study did not identify any off-target binding for NI301A at 30 µg/mL, the highest concentration tested. No staining was observed on liver tissue sections, the main organ producing TTR, further confirming NI301A absent binding to physiological TTR protein (supplementary Fig. 4)."

Minor concerns:

4. Is the NI006 antibody currently used in a ATTR cardiomyopathy clinic trial identical to the NI301A antibody?

Response:

NI301A and NI006 are two successive names for the same antibody. NI006 depicts the clinical development candidate. Both names are now referenced in the introduction and discussion sections of the manuscript.

5. The scale bar is missing in Figure 4 B3. The size of the scale bar is missing or not visible or provided in the remainder of images with IF/IHC staining.

Response:

We thank the reviewer for spotting this error. Adequately sized scale bars and text were added to each image.

6. The protein size in Figure 1H should be indicated or a size marker should be provided. Is this the only protein band after immunoprecipitation with human serum?

Response:

The protein size marker is now indicated in Figure 1H. We have also complemented this figure with the isotype control antibody that was run at the same time. The 14 kDa band shown in this figure corresponds to TTR monomers and is the only relevant band present on the membranes.

Complete pictures of the western blot membranes have been added as a supplementary figure 2. The following bands are visible on the membrane: TTR monomers (14 kDa), TTR dimers (30 kDa) and human heavy chains (50 kDa). The latter are detected by the HRP-conjugated Protein A used for detection and could be present as carry-over from human serum or from the NI301A and isotype control antibodies that were used for IP.

7. Please double check the bar size of Figure 2A, A - control of DAKO A0002 staining image. It appears to use a different magnification when compared to the others, but an identical scale bar is given.

Response:

We confirm that this image (Figure 2A, A- Control , DAKO A0002) is at the same scale than the other images in this panel. (The scale bar present within each image is generated automatically by the digital slide scanner).

The reason for the different appearance of this tissue, which may indeed give the impression that the magnification would be different, relates to the different processing of the tissues and resulting tissue quality. While NI301A IHC and congo red staining and birefringence were performed on fresh-frozen tissue blocks, staining with Dako A0002 does not work on fresh frozen tissues. The latter therefore required the use of a second set of tissue blocks from the same organ donors which were thawed, post-fixed in PFA and embedded in paraffin. Possibly due to a fixation related tissue shrinkage, tissues used for Dako A0002 IHC present a different appearance than the ones used for NI301A IHC and CR staining.

We have highlighted the different procedures in the Material and Methods as follows:

"NI301A was used as a biotinylated conjugate at 10 nM on frozen tissue sections. Antibody Dako A0002 was used at 1:500-1:1000 dilutions and required PFA post-fixation for staining, leading to a different appearance of the tissues due to the different processing."

8. How many replicate experiments have been performed in Figure 4B1 and B2?

Response:

The experiment reported in Figure 4B1 and B2 was performed twice. Both experiments delivered similar results, and only one representative experiment was included in the manuscript.

Reviewer #2 (Remarks to the Author):

This study represents to the development of an antibody that is specific only for miss folded transthyretin and does not bind to transthyretin in the native tetra her state. The binding is specific to the Fc fragment of the immunoglobulin and is enhance by the use of complement.

The antibody binds to 5 amino acids that are not exposed in native transthyretin but only exposed when miss folded both in mutant as well as wild-type amyloid deposits. The controls used in this trial are appropriate in the antibody was tested in multiple different mutations of amyloid.

In vitro binding occurs with the antibody were commercial anti transthyretin anti serum fails to bind. The antibody can also buying to amyloid deposits placed in mice and triggers phagocytosis resulting in resolution of the deposits. 9/80 autopsy he has for heart failure patients over the age of 80 demonstrated ATTR deposits suggesting this is a widespread problem for which this antibody offers a potential solution beyond the currently available stabilizing therapies.

9. The only question I have is whether any toxicity studies were done on the mice that had fibro graft placement followed by injection of monoclonal mouse chimeric antibody.

Response:

In the xenograft model, mice did not exhibit any signs of overt toxicity based on daily animal observation. This information has been added to the results and to the discussion.

“At this late time point (96 hours), ATTR fibril grafts were entirely infiltrated by phagocytic monocytes (CD11B+, LY6G-, IBA1) (supplementary Fig. 5), and mice did not exhibit any sign of overt toxicity.”

“A murinized chimeric variant of NI301A substantially accelerated the removal of ATTR by phagocytic cells and did not exhibit any signs of overt toxicity in this model.”

To further assess the safety profile of NI301A, a GLP toxicity study was conducted in rats in compliance with regulatory guidelines. In this study, NI301A did not elicit any detectable toxicity upon repeated administrations at 300 mg/kg, the highest dose investigated. This toxicity study will be reported in an independent publication focusing on the non-clinical evaluation of NI006.

Reviewer #3 (Remarks to the Author):

In their paper “A human antibody selective for transthyretin amyloid removes cardiac amyloid through phagocytic immune cells” Michalon et al present data on a new antibody therapy for Transthyretin amyloid cardiomyopathy, a rare condition which is characterised by the deposition of extracellular fibrils in the myocardium. Reducing myocardial fibrils load as a therapeutic goal has been demonstrated before which reduces the novelty of the work. The authors present NI301A, a monoclonal antibody obtained from memory B cell of healthy elderly subjects.

The paper is well written, the data backs up the claims made and the authors present a large amount of in vitro and in vivo data. The demonstrate that the antibody can dissolve fibrils in vitro (well controlled with heat inactivated plasma etc) and in vivo using an elegant patient derived xenografts model with a purpose made human/mouse chimeric antibody.

Overall, this is a strong and important study tackling a rare but life threatening disease.

My main comment is that whilst the authors discuss the use of other approaches such as small molecule therapeutics for the indication e.g. tafamidis and diflunisal as well as gene therapeutics patisiran and inotersen they ignore other therapeutics approaches such as Dezamizumab. To provide convincing data that the selected antibody approach is superior in reversing existing disease rather than stopping progression it is essential to do a head-to-head comparison in vitro but most importantly in vivo with competing antibodies and perhaps also other approach currently under investigation.

Response:

To address the reviewer’s comment, we have included the evaluation of tafamidis activity in the xenograft model. Tafamidis is currently the only approved therapy for ATTR-CM. The fibril xenograft model has been developed to evaluate a treatment effect to reverse existing disease (i.e. existing amyloid load), in contrast to other models which are directed towards amyloid formation. Our results indicated that ch.NI301A fibril removal activity was maintained in presence of tafamidis, and that tafamidis alone had no effect on ATTR fibril removal.

We have added the corresponding results as figure 6F, complemented the material and method sections and added the following text to the manuscript and to the figure legend: *“The effect of ch.NI301A was also evaluated in presence of the TTR stabilizer tafamidis, administered daily at 1 mg/kg i.p. ch.NI301A fibril removal activity was maintained in presence of tafamidis. Tafamidis alone had no effect on ATTR fibril removal (Fig. 6F).”*

“F) Evaluation of tafamidis alone and ch.NI301A+tafamidis combination on ATTR removal in vivo (mean \pm sd of n=6-8 mice per group, 3-4 sections per mouse).”

Clinical development of dezamizumab was terminated in 2018 due to an apparent change in the risk-benefit profile. We have included the following paragraph on dezamizumab to the discussion:

“Dezamizumab is a humanized monoclonal IgG1 antibody binding to native Serum Amyloid P component, a plasma protein identified in all types of amyloid deposits. In a phase 1 clinical trial conducted in patients with systemic amyloidosis, dezamizumab treatment was associated with signs of organ response such as decreases in liver stiffness which were highly consistent with the macrophage-dependent phagocytic clearance of amyloid demonstrated in animal models. A phase II trial in patients with cardiac amyloidosis was initiated but terminated prematurely due to an apparent change in the risk-benefit profile, which might have been related to dezamizumab binding physiological SAP protein.”

REVIEWERS' COMMENTS

Reviewer #1 (Remarks to the Author):

The manuscript has been suitably revised and is now acceptable for publication.

Reviewer #2 (Remarks to the Author):

none

Reviewer #3 (Remarks to the Author):

The authors have performed additional experiments to strengthen the manuscript. I have no further comments.

Nature Communications manuscript NCOMMS-20-34138-T:

"A human antibody selective for transthyretin amyloid removes cardiac amyloid through phagocytic immune cells"

RESPONSE TO REFEREES

Reviewer #1 (Remarks to the Author):

The manuscript has been suitably revised and is now acceptable for publication.

Reviewer #2 (Remarks to the Author):

none

Reviewer #3 (Remarks to the Author):

The authors have performed additional experiments to strengthen the manuscript. I have no further comments.

Response:

We thank the reviewers for their constructive comments and their appreciation of the manuscript.